# Universal Style Transfer via Feature Transforms

**Yijun Li**
UC Merced
yli62@ucmerced.edu

**Chen Fang**
Adobe Research
cfang@adobe.com

**Jimei Yang**
Adobe Research
jimyang@adobe.com

**Zhaowen Wang**
Adobe Research
zhawang@adobe.com

**Xin Lu**
Adobe Research
xinl@adobe.com

**Ming-Hsuan Yang**
UC Merced, NVIDIA Research
mhyang@ucmerced.edu

## Abstract

Universal style transfer aims to transfer arbitrary visual styles to content images. Existing feed-forward based methods, while enjoying the inference efficiency, are mainly limited by inability of generalizing to unseen styles or compromised visual quality. In this paper, we present a simple yet effective method that tackles these limitations without training on any pre-defined styles. The key ingredient of our method is a pair of feature transforms, whitening and coloring, that are embedded to an image reconstruction network. The whitening and coloring transforms reflect a direct matching of feature covariance of the content image to a given style image, which shares similar spirits with the optimization of Gram matrix based cost in neural style transfer. We demonstrate the effectiveness of our algorithm by generating high-quality stylized images with comparisons to a number of recent methods. We also analyze our method by visualizing the whitened features and synthesizing textures via simple feature coloring.

## 1 Introduction

Style transfer is an important image editing task which enables the creation of new artistic works. Given a pair of examples, i.e., the content and style image, it aims to synthesize an image that preserves some notion of the content but carries characteristics of the style. The key challenge is how to extract effective representations of the style and then match it in the content image. The seminal work by Gatys et al. [8, 9] show that the correlation between features, i.e., Gram matrix or covariance matrix (shown to be as effective as Gram matrix in [20]), extracted by a trained deep neural network has remarkable ability of capturing visual styles. Since then, significant efforts have been made to synthesize stylized images by minimizing Gram/covariance matrices based loss functions, through either iterative optimization [9] or trained feed-forward networks [27, 16, 20, 2, 6]. Despite the recent rapid progress, these existing works often trade off between generalization, quality and efficiency, which means that optimization-based methods can handle arbitrary styles with pleasing visual quality but at the expense of high computational costs, while feed-forward approaches can be executed efficiently but are limited to a fixed number of styles or compromised visual quality.

By far, the problem of universal style transfer remains a daunting task as it is challenging to develop neural networks that achieve generalization, quality and efficiency at the same time. The main issue is how to properly and effectively apply the extracted style characteristics (feature correlations) to content images in a style-agnostic manner.

In this work, we propose a simple yet effective method for universal style transfer, which enjoys the style-agnostic generalization ability with marginally compromised visual quality and execution efficiency. The transfer task is formulated as image reconstruction processes, with the content features

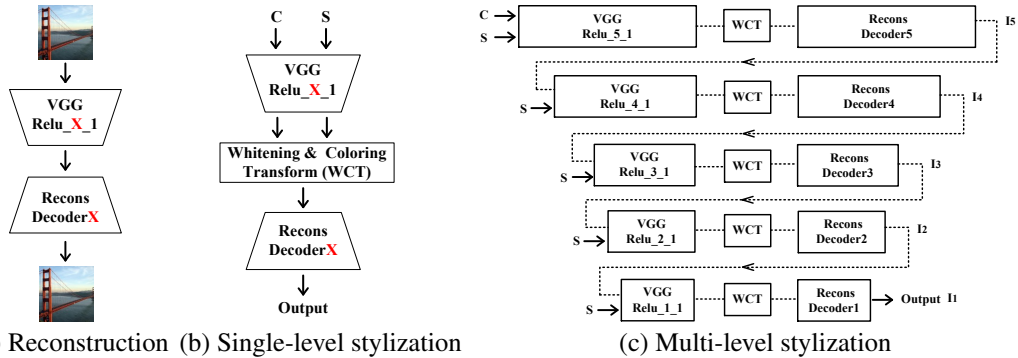

(a) Reconstruction (b) Single-level stylization          (c) Multi-level stylization

Figure 1: Universal style transfer pipeline. (a) We first pre-train five decoder networks DecoderX (X=1,2,...,5) through image reconstruction to invert different levels of VGG features. (b) With both VGG and DecoderX *fixed*, and given the content image $C$ and style image $S$, our method performs the style transfer through whitening and coloring transforms. (c) We extend single-level to multi-level stylization in order to match the statistics of the style at all levels. The result obtained by matching higher level statistics of the style is treated as the new content to continue to match lower-level information of the style.

being *transformed* at intermediate layers with regard to the statistics of the style features, in the midst of feed-forward passes. In each intermediate layer, our main goal is to transform the extracted content features such that they exhibit the same statistical characteristics as the style features of the same layer and we found that the classic signal *whitening* and *coloring* transforms (WCTs) on those features are able to achieve this goal in an almost effortless manner.

In this work, we first employ the VGG-19 network [26] as the feature extractor (encoder), and train a symmetric decoder to invert the VGG-19 features to the original image, which is essentially the image reconstruction task (Figure 1(a)). Once trained, both the encoder and the decoder are fixed through all the experiments. To perform style transfer, we apply WCT to one layer of content features such that its covariance matrix matches that of style features, as shown in Figure 1(b). The transformed features are then fed forward into the downstream decoder layers to obtain the stylized image. In addition to this single-level stylization, we further develop a multi-level stylization pipeline, as depicted in Figure 1(c), where we apply WCT sequentially to multiple feature layers. The multi-level algorithm generates stylized images with greater visual quality, which are comparable or even better with much less computational costs. We also introduce a control parameter that defines the degree of style transfer so that the users can choose the balance between stylization and content preservation. The entire procedure of our algorithm only requires learning the image reconstruction decoder with *no* style images involved. So when given a new style, we simply need to extract its feature covariance matrices and apply them to the content features via WCT. Note that this learning-free scheme is fundamentally different from existing feed-forward networks that require learning with pre-defined styles and fine-tuning for new styles. Therefore, our approach is able to achieve style transfer universally.

The main contributions of this work are summarized as follows:

- We propose to use feature transforms, i.e., whitening and coloring, to directly match content feature statistics to those of a style image in the deep feature space.
- We couple the feature transforms with a pre-trained general encoder-decoder network, such that the transferring process can be implemented by simple feed-forward operations.
- We demonstrate the effectiveness of our method for universal style transfer with high-quality visual results, and also show its application to universal texture synthesis.

## 2 Related Work

Existing style transfer methods are mostly example-based [13, 25, 24, 7, 21]. The image analogy method [13] aims to determine the relationship between a pair of images and then apply it to stylize

other images. As it is based on finding dense correspondence, analogy-based approaches [25, 24, 7, 21] often require that a pair of image depicts the same type of scene. Therefore these methods do not scale to the setting of arbitrary style images well.

Recently, Gatys et al. [8, 9] proposed an algorithm for arbitrary stylization based on matching the correlations (Gram matrix) between deep features extracted by a trained network classifier within an iterative optimization framework. Numerous methods have since been developed to address different aspects including speed [27, 19, 16], quality [28, 18, 32, 31], user control [10], diversity [29, 20], semantics understanding [7, 1] and photorealism [23]. It is worth mentioning that one of the major drawbacks of [8, 9] is the inefficiency due to the optimization process. The improvement of efficiency in [27, 19, 16] is realized by formulating the stylization as learning a feed-forward image transformation network. However, these methods are limited by the requirement of training one network per style due to the lack of generalization in network design.

Most recently, a number of methods have been proposed to empower a single network to transfer multiple styles, including a model that conditioned on binary selection units [20], a network that learns a set of new filters for every new style [2], and a novel conditional normalization layer that learns normalization parameters for each style [6]. To achieve arbitrary style transfer, Chen et al. [3] first propose to swap the content feature with the closest style feature locally. Meanwhile, inspired by [6], two following work [30, 11] turn to learn a general mapping from the style image to style parameters. One closest related work [15] directly adjusts the content feature to match the mean and variance of the style feature. However, the generalization ability of the learned models on unseen styles is still limited.

Different from the existing methods, our approach performs style transfer efficiently in a feed-forward manner while achieving generalization and visual quality on arbitrary styles. Our approach is closely related to [15], where content feature in a particular (higher) layer is adaptively instance normalized by the mean and variance of style feature. This step can be viewed as a sub-optimal approximation of the WCT operation, thereby leading to less effective results on both training and unseen styles. Moreover, our encoder-decoder network is trained solely based on image reconstruction, while [15] requires learning such a module particularly for stylization task. We evaluate the proposed algorithm with existing approaches extensively on both style transfer and texture synthesis tasks and present in-depth analysis.

## 3 Proposed Algorithm

We formulate style transfer as an image reconstruction process coupled with feature transformation, i.e., whitening and coloring. The reconstruction part is responsible for inverting features back to the RGB space and the feature transformation matches the statistics of a content image to a style image.

### 3.1 Reconstruction decoder

We construct an auto-encoder network for general image reconstruction. We employ the VGG-19 [26] as the encoder, fix it and train a decoder network simply for inverting VGG features to the original image, as shown in Figure 1(a). The decoder is designed as being symmetrical to that of VGG-19 network (up to Relu_X_1 layer), with the nearest neighbor upsampling layer used for enlarging feature maps. To evaluate with features extracted at different layers, we select feature maps at five layers of the VGG-19, i.e., Relu_X_1 (X=1,2,3,4,5), and train five decoders accordingly. The pixel reconstruction loss [5] and feature loss [16, 5] are employed for reconstructing an input image,

$$L = \|I_o - I_i\|_2^2 + \lambda\|\Phi(I_o) - \Phi(I_i)\|_2^2 \,, \tag{1}$$

where $I_i$, $I_o$ are the input image and reconstruction output, and $\Phi$ is the VGG encoder that extracts the Relu_X_1 features. In addition, $\lambda$ is the weight to balance the two losses. After training, the decoder is fixed (i.e., will not be fine-tuned) and used as a feature inverter.

### 3.2 Whitening and coloring transforms

Given a pair of content image $I_c$ and style image $I_s$, we first extract their vectorized VGG feature maps $f_c \in \Re^{C \times H_c W_c}$ and $f_s \in \Re^{C \times H_s W_s}$ at a certain layer (e.g., Relu_4_1), where $H_c$, $W_c$ ($H_s$,

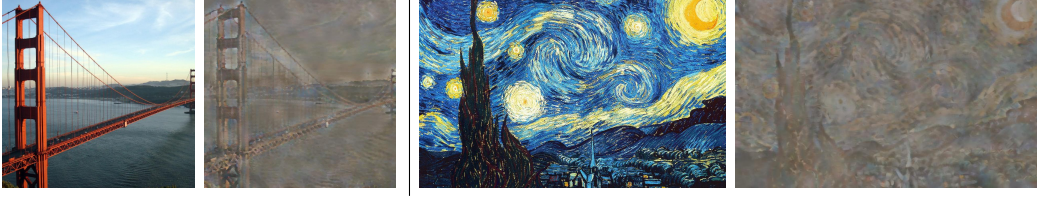

Figure 2: Inverting whitened features. We invert the whitened VGG Relu_4_1 feature as an example. Left: original images, Right: inverted results (pixel intensities are rescaled for better visualization). The whitened features still maintain global content structures.

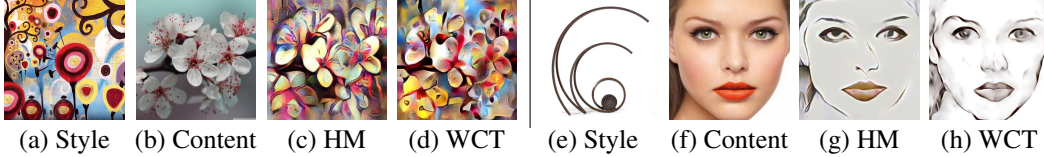

(a) Style   (b) Content   (c) HM   (d) WCT   (e) Style   (f) Content   (g) HM   (h) WCT

Figure 3: Comparisons between different feature transform strategies. Results are obtained by our multi-level stylization framework in order to match all levels of information of the style.

$W_s$) are the height and width of the content (style) feature, and $C$ is the number of channels. The decoder will reconstruct the original image $I_c$ if $f_c$ is directly fed into it. We next propose to use a whitening and coloring transform to adjust $f_c$ with respect to the statistics of $f_s$. The goal of WCT is to directly transform the $f_c$ to match the covariance matrix of $f_s$. It consists of two steps, i.e., whitening and coloring transform.

**Whitening transform.** Before whitening, we first center $f_c$ by subtracting its mean vector $m_c$. Then we transform $f_c$ linearly as in (2) so that we obtain $\hat{f}_c$ such that the feature maps are uncorrelated $(\hat{f}_c \hat{f}_c^\top = I)$,

$$\hat{f}_c = E_c \, D_c^{-\frac{1}{2}} \, E_c^\top \, f_c \,, \tag{2}$$

where $D_c$ is a diagonal matrix with the eigenvalues of the covariance matrix $f_c \, f_c^\top \in \Re^{C \times C}$, and $E_c$ is the corresponding orthogonal matrix of eigenvectors, satisfying $f_c \, f_c^\top = E_c D_c E_c^\top$.

To validate what is encoded in the whitened feature $\hat{f}_c$, we invert it to the RGB space with our previous decoder trained for reconstruction only. Figure 2 shows two visualization examples, which indicate that the whitened features still maintain global structures of the image contents, but greatly help remove other information related to styles. We note especially that, for the *Starry_night* example on right, the detailed stroke patterns across the original image are gone. In other words, the whitening step helps peel off the style from an input image while preserving the global content structure. The outcome of this operation is ready to be transformed with the target style.

**Coloring transform.** We first center $f_s$ by subtracting its mean vector $m_s$, and then carry out the coloring transform [14], which is essentially the inverse of the whitening step to transform $\hat{f}_c$ linearly as in (3) such that we obtain $\hat{f}_{cs}$ which has the desired correlations between its feature maps $(\hat{f}_{cs} \hat{f}_{cs}^\top = f_s \, f_s^\top)$,

$$\hat{f}_{cs} = E_s \, D_s^{\frac{1}{2}} \, E_s^\top \, \hat{f}_c \,, \tag{3}$$

where $D_s$ is a diagonal matrix with the eigenvalues of the covariance matrix $f_s \, f_s^\top \in \Re^{C \times C}$, and $E_s$ is the corresponding orthogonal matrix of eigenvectors. Finally we re-center the $\hat{f}_{cs}$ with the mean vector $m_s$ of the style, i.e., $\hat{f}_{cs} = \hat{f}_{cs} + m_s$.

To demonstrate the effectiveness of WCT, we compare it with a commonly used feature adjustment technique, i.e., histogram matching (HM), in Figure 3. The channel-wise histogram matching [12] method determines a mapping function such that the mapped $f_c$ has the same cumulative histogram as $f_s$. In Figure 3, it is clear that the HM method helps transfer the global color of the style image well

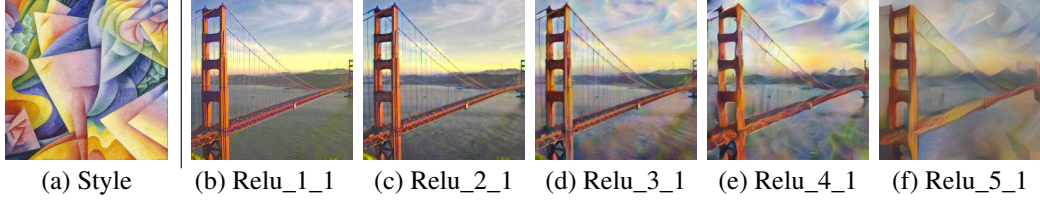

|(a) Style|(b) Relu_1_1|(c) Relu_2_1|(d) Relu_3_1|(e) Relu_4_1|(f) Relu_5_1|

Figure 4: Single-level stylization using different VGG features. The content image is from Figure 2.

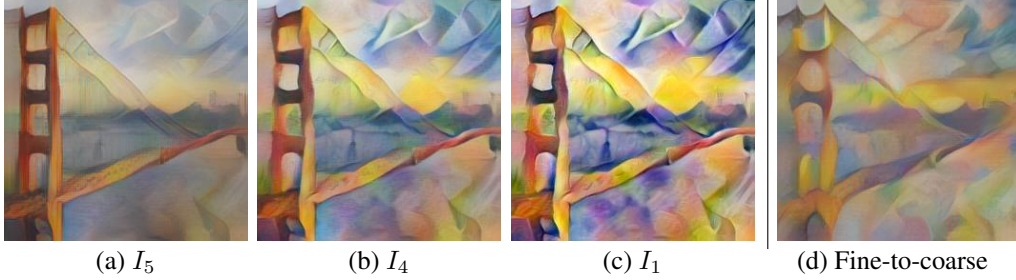

| (a) $I_5$ | (b) $I_4$ | (c) $I_1$ | (d) Fine-to-coarse |

Figure 5: (a)-(c) Intermediate results of our coarse-to-fine multi-level stylization framework in Figure 1(c). The style and content images are from Figure 4. $I_1$ is the final output of our multi-level pipeline. (d) Reversed fine-to-coarse multi-level pipeline.

but fails to capture salient visual patterns, e.g., patterns are broken into pieces and local structures are misrepresented. In contrast, our WCT captures patterns that reflect the style image better. This can be explained by that the HM method does not consider the correlations between features channels, which are exactly what the covariance matrix is designed for.

After the WCT, we may blend $\hat{f_{cs}}$ with the content feature $f_c$ as in (4) before feeding it to the decoder in order to provide user controls on the strength of stylization effects:

$$\hat{f_{cs}} = \alpha \, \hat{f_{cs}} + (1 - \alpha) \, f_c \, , \tag{4}$$

where $\alpha$ serves as the style weight for users to control the transfer effect.

### 3.3 Multi-level coarse-to-fine stylization

Based on the single-level stylization framework shown in Figure 1(b), we use different layers of VGG features Relu_X_1 (X=1,2,...,5) and show the corresponding stylized results in Figure 4. It clearly shows that the higher layer features capture more complicated local structures, while lower layer features carry more low-level information (e.g., colors). This can be explained by the increasing size of receptive field and feature complexity in the network hierarchy. Therefore, it is advantageous to use features at all five layers to fully capture the characteristics of a style from low to high levels.

Figure 1(c) shows our multi-level stylization pipeline. We start by applying the WCT on Relu_5_1 features to obtain a coarse stylized result and regard it as the new content image to further adjust features in lower layers. An example of intermediate results are shown in Figure 5. We show the intermediate results $I_5$, $I_4$, $I_1$ with obvious differences, which indicates that the higher layer features first capture salient patterns of the style and lower layer features further improve details. If we reverse feature processing order (i.e., fine-to-coarse layers) by starting with Relu_1_1, low-level information cannot be preserved after manipulating higher level features, as shown in Figure 5(d).

## 4 Experimental Results

### 4.1 Decoder training

For the multi-level stylization approach, we separately train five reconstruction decoders for features at the VGG-19 Relu_X_1 (X=1,2,...,5) layer. It is trained on the Microsoft COCO dataset [22] and the weight $\lambda$ to balance the two losses in (1) is set as 1.

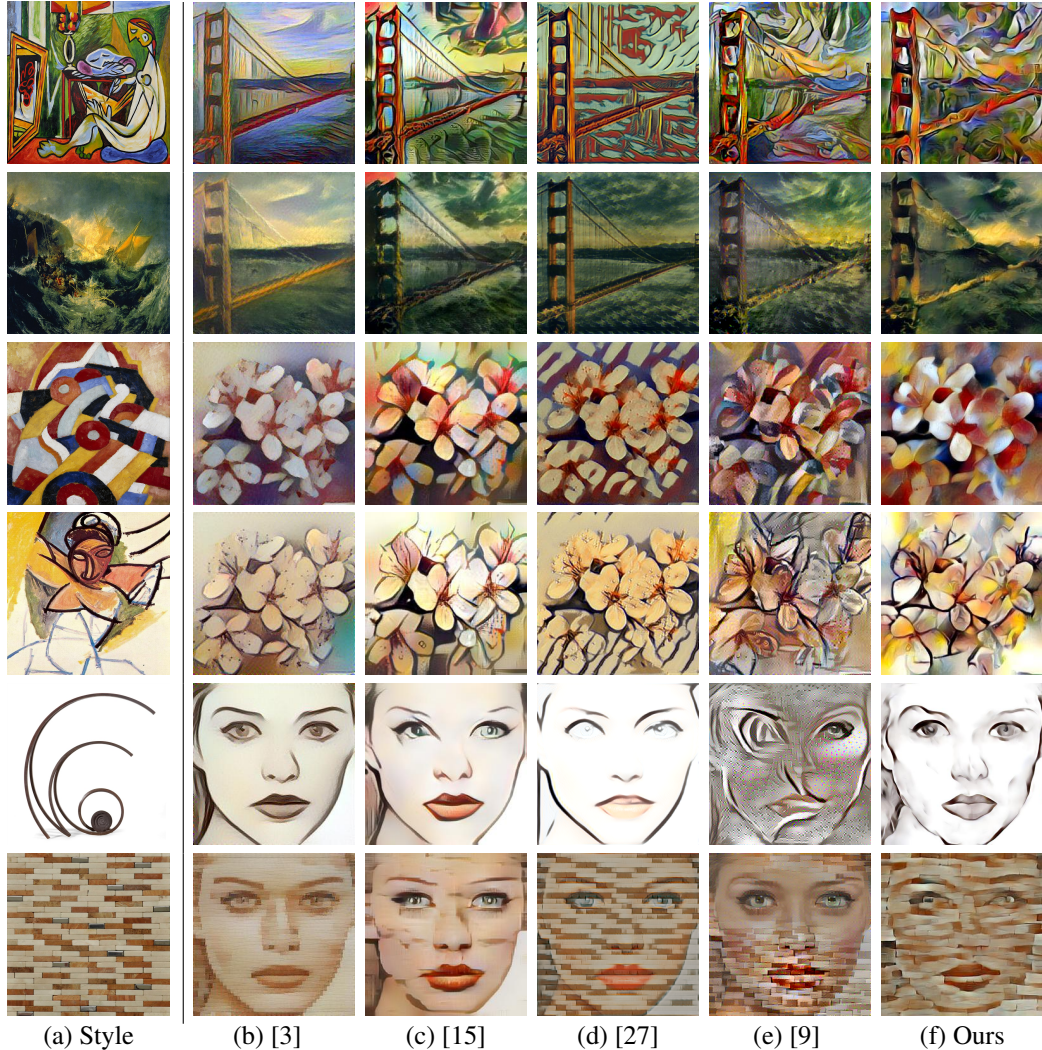

| (a) Style | (b) [3] | (c) [15] | (d) [27] | (e) [9] | (f) Ours |

Figure 6: Results from different style transfer methods. The content images are from Figure 2-3. We evaluate various styles including paintings, abstract styles, and styles with obvious texton elements. We adjust the style weight of each method to obtain the best stylized effect. For our results, we set the style weight $\alpha = 0.6$.

Table 1: Differences between our approach and other methods.

|  | Chen et al. [3] | Huang et al. [15] | TNet [27] | DeepArt [9] | Ours |
|---|---|---|---|---|---|
| Arbitrary | √ | √ | × | √ | √ |
| Efficient | √ | √ | √ | × | √ |
| Learning-free | × | × | × | √ | √ |

## 4.2 Style transfer

To demonstrate the effectiveness of the proposed algorithm, we list the differences with existing methods in Table 1 and present stylized results in Figure 6. We adjust the style weight of other methods to obtain the best stylized effect. The optimization-based work of [9] handles arbitrary styles but is likely to encounter unexpected local minima issues (e.g., 5th and 6th row of Figure 6(e)). Although the method [27] greatly improves the stylization speed, it trades off quality and generality for efficiency, which generates repetitive patterns that overlay with the image contents (Figure 6(d)).

Table 2: Quantitative comparisons between different stylization methods in terms of the covariance matrix difference ($L_s$), user preference and run-time, tested on images of size $256 \times 256$ and a 12GB TITAN X.

|  | Chen et al. [3] | Huang et al. [15] | TNet [27] | Gatys et al. [9] | Ours |
|---|---|---|---|---|---|
| $\log(L_s)$ | 7.4 | 7.0 | 6.8 | 6.7 | **6.3** |
| Preference/% | 15.7 | 24.9 | 12.7 | 16.4 | **30.3** |
| Time/sec | 2.1 | 0.20 | 0.18 | 21.2 | **0.83** |

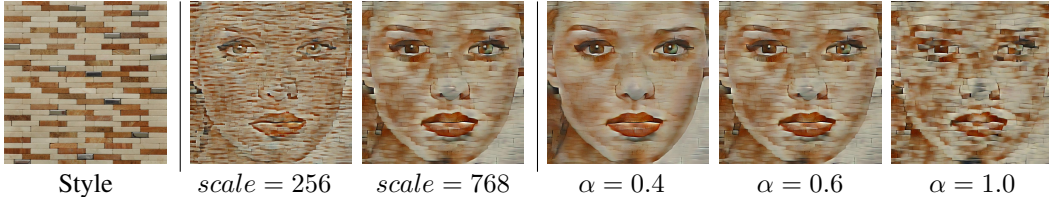

| Style | $scale = 256$ | $scale = 768$ | $\alpha = 0.4$ | $\alpha = 0.6$ | $\alpha = 1.0$ |

Figure 7: Controlling the stylization on the scale and weight.

Closest to our work on generalization are the recent methods [3, 15], but the quality of the stylized results are less appealing. The work of [3] replaces the content feature with the most similar style feature based on patch similarity and hence has limited capability, i.e., the content is strictly preserved while style is not well reflected with only low-level information (e.g., colors) transferred, as shown in Figure 6(b). In [15], the content feature is simply adjusted to have the same mean and variance with the style feature, which is not effective in capturing high-level representations of the style. Even learned with a set of training styles, it does not generalize well on unseen styles. Results in Figure 6(c) indicate that the method in [15] is not effective at capturing and synthesizing salient style patterns, especially for complicated styles where there are rich local structures and non-smooth regions.

Figure 6(f) shows the stylized results of our approach. Without learning any style, our method is able to capture visually salient patterns in style images (e.g., the brick wall on the 6th row). Moreover, key components in the content images (e.g., bridge, eye, mouth) are also well stylized in our results, while other methods only transfer patterns to relatively smooth regions (e.g., sky, face). The models and code are available at `https://github.com/Yijunmaverick/UniversalStyleTransfer`.

In addition, we quantitatively evaluate different methods by computing the covariance matrix difference ($L_s$) on all five levels of VGG features between stylized results and the given style image. We randomly select 10 content images from [22] and 40 style images from [17], compute the averaged difference over all styles, and show the results in Table 2 (1st row). Quantitative results show that we generate stylized results with lower $L_s$, i.e., closer to the statistics of the style.

**User study.** Evaluating artistic style transfer has been an open question in the community. Since the qualitative assessment is highly subjective, we conduct a user study to evaluate 5 methods shown in Figure 6. We use 5 content images and 30 style images, and generate 150 results based on each content/style pair for each method. We randomly select 15 style images for each subject to evaluate. We display stylized images by 5 compared methods side-by-side on a webpage in random order. Each subject is asked to vote his/her ONE favorite result for each style. We finally collect the feedback from 80 subjects of totally 1,200 votes and show the percentage of the votes each method received in Table 2 (2nd row). The study shows that our method receives the most votes for better stylized results. It can be an interesting direction to develop evaluation metrics based on human visual perception for general image synthesis problems.

**Efficiency.** In Table 2 (3rd row), we also compare our approach with other methods in terms of efficiency. The method by Gatys et al. [9] is slow due to loops of optimization and usually requires at least 500 iterations to generate good results. The methods [27] and [15] are efficient as the scheme is based on one feed-forward pass with a trained network. The approach [3] is feed-forward based but relatively slower as the feature swapping operation needs to be carried out for thousands of patches. Our approach is also efficient but a little bit slower than [27, 15] because we have a eigenvalue decomposition step in WCT. But note that the computational cost on this step will not increase along with the image size because the the dimension of covariance matrix only depends on filter numbers (or

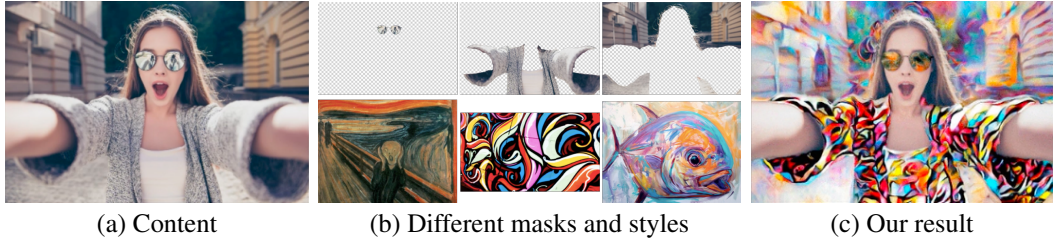

(a) Content        (b) Different masks and styles        (c) Our result

Figure 8: Spatial control in transferring, which enables users to edit the content with different styles.

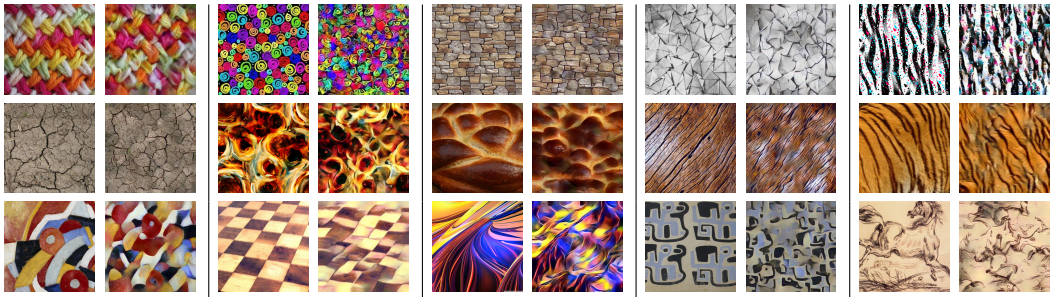

Figure 9: Texture synthesis. In each panel, Left: original textures, Right: our synthesized results. Texture images are mostly from the Describable Textures Dataset (DTD) [4].

channels), which is at most 512 (Relu_5_1). Currently the decomposition step is implemented based on CPU. Our future work includes more efficient GPU implementations of the proposed algorithm.

**User Controls.** Given a content/style pair, our approach is not only as simple as a one-click transferring, but also flexible enough to accommodate different requirements from users by providing different controls on the stylization, including the scale, weight and spatial control. The style input on different scales will lead to different extracted statistics due to the fixed receptive field of the network. Therefore the scale control is easily achieved by adjusting the style image size. In the middle of Figure 7, we show two examples where the brick can be transferred in either small or large scale. The weight control refers to controlling the balance between stylization and content preservation. As shown on right of Figure 7, our method enjoys this flexibility in simple feed-forward passes by simply adjusting the style weight $\alpha$ in (4). However in [9] and [27], to obtain visual results of different weight settings, a new round of time-consuming optimization or model training is needed. Moreover, our blending directly works on deep feature space before inversion/reconstruction, which is fundamentally different from [9, 27] where the blending is formulated as the weighted sum of the content and style losses that may not always lead to a good balance point.

The spatial control is also highly desired when users want to edit an image with different styles transferred on different parts of the image. Figure 8 shows an example of spatially controlling the stylization. A set of masks $M$ (Figure 8(b)) is additionally required as input to indicate the spatial correspondence between content regions and styles. By replacing the content feature $f_c$ in (3) with $M \odot f_c$ where $\odot$ is a simple mask-out operation, we are able to stylize the specified region only.

### 4.3 Texture synthesis

By setting the content image as a random noise image (e.g., Gaussian noise), our stylization framework can be easily applied to texture synthesis. An alternative is to directly initialize the $\hat{f}_c$ in (3) to be white noise. Both approaches achieve similar results. Figure 9 shows a few examples of the synthesized textures. We empirically find that it is better to run the multi-level pipeline for a few times (e.g., 3) to get more visually pleasing results.

Our method is also able to synthesize the interpolated result of two textures. Given two texture examples $s_1$ and $s_2$, we first perform the WCT on the input noise and get transformed features $\hat{f}_{cs_1}$ and $\hat{f}_{cs_2}$ respectively. Then we blend these two features $\hat{f}_{cs} = \beta \hat{f}_{cs_1} + (1 - \beta) \hat{f}_{cs_2}$ and feed the

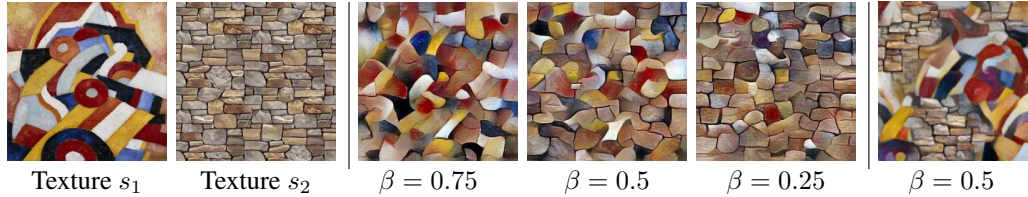

| Texture $s_1$ | Texture $s_2$ | $\beta = 0.75$ | $\beta = 0.5$ | $\beta = 0.25$ | $\beta = 0.5$ |

Figure 10: Interpolation between two texture examples. Left: original textures, Middle: our interpolation results, Right: interpolated results of [9]. $\beta$ controls the weight of interpolation.

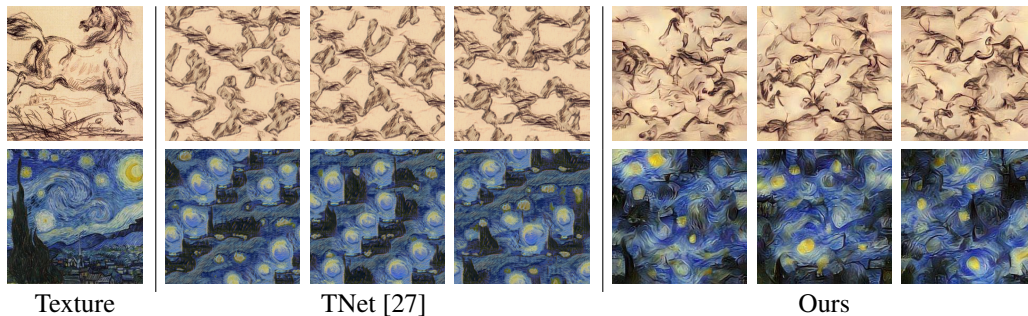

Texture            TNet [27]                    Ours

Figure 11: Comparisons of diverse synthesized results between TNet [27] and our model.

combined feature into the decoder to generate mixed effects. Note that our interpolation directly works on deep feature space. By contrast, the method in [9] generates the interpolation by matching the weighted sum of Gram matrices of two textures at the loss end. Figure 10 shows that the result by [9] is simply overlaid by two textures while our method generates new textural effects, e.g., bricks in the stripe shape.

One important aspect in texture synthesis is *diversity*. By sampling different noise images, our method can generate diverse synthesized results for each texture. While [27] can generate different results driven by the input noise, the learned networks are very likely to be trapped in local optima. In other words, the noise is marginalized out and thus fails to drive the network to generate large visual variations. In contrast, our approach explains each input noise better because the network is unlikely to absorb the variations in input noise since it is never trained for learning textures. We compare the diverse outputs of our model with [27] in Figure 11. Note that the common diagonal layout is shared across different results of [27], which causes unsatisfying visual experiences. The comparison shows that our method achieves diversity in a more natural and flexible manner.

## 5   Concluding Remarks

In this work, we propose a universal style transfer algorithm that does not require learning for each individual style. By unfolding the image generation process via training an auto-encoder for image reconstruction, we integrate the whitening and coloring transforms in the feed-forward passes to match the statistical distributions and correlations between the intermediate features of content and style. We also present a multi-level stylization pipeline, which takes all level of information of a style into account, for improved results. In addition, the proposed approach is shown to be equally effective for texture synthesis. Experimental results demonstrate that the proposed algorithm achieves favorable performance against the state-of-the-art methods in generalizing to arbitrary styles.

**Acknowledgments**

This work is supported in part by the NSF CAREER Grant #1149783, gifts from Adobe and NVIDIA.

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
