[Reviews · NeurIPS 2017]

Reviewer 1



The authors propose a style transfer algorithm that is universal to styles (need not train a new style model for different styles). The main contributions as authors pointed out are: 1. Using whitening and color transform (WCT), 2) using a encoder-decoder architecture and VGG model for style adaptation making it purely feed-forward. There are a lot of previous works on style transfer that explicitly train style specific models so a universal model is better and while there have been a few works in universal style transfer models this particular work on using WCT is interesting. While the authors did compare to many methods it is not absolutely clear that this is the best method for universal style transfer because of speed (some other methods are faster) and because there is no qualitative study. I still want to encourage the paper as is. I do suggest the authors push for a small A/B test to really understand the qualitative gains the proposed method has achieved.

Reviewer 2



The paper presents a very simple idea for performing style transfer: simply use a normal auto-encoder, whiten the features of the content image and color them with the statistics from the features of the style image. It's well written, the idea are is presented clearly and the evaluation is as good as can be expected on style transfer. I know a few works in this area, but I'm not following precisely the latest developments, and there has been a very high number of paper on this topic in the last year, so it is hard for me to evaluate the novelty and interest of this paper. Technically, it seems to me a marginal improvement over [14]. In term of results, I find style transfer work extremely difficult to evaluate and this paper was not an exception: it's clearly close to the state of the art, it's not clear to me it's really better than it's competitors. This alone would lead me to a weak reject. The reason I would rather tend to support the paper is because I have the impression the approach shows something about the way information is encoded in CNN features that goes beyond what can be said from Gatys et al. (and I think that's the reason why nobody - as far as I know - has tried this simple method yet) : indeed, there is nothing in the presented method that forces the content of the images to be the same, it could potentially be completely destroyed by the whitening/coloring. This aspect is however not explored much in the paper: - the standard type of content loss to table 1 (feature reconstruction of high level features) - fig 7e worries me a bit, it seems all the content has really disappeared: I would like to see much more of this image, with different styles. If it appear this is a general case, I would want a good justification of why there is more than interpolation in a good feature space going on and/or why the feature space of the auto-encoder is good for style transfer. If, as I suspect, it's not the general case, it would be nice to see it. - in general, I think the work would be much more interesting if it focused more on the analysis of the way information is encoded in the representation. Figure 2 and 3 are a bit in this direction, but it would be good to have more and also than images (e.g. linear classification performances). I think there would be another point to consider for acceptance: NIPS is not really my community but I wonder if NIPS is really a good venue for this type of applied work, with little theory and little possibility of (quantitative) evaluation.

Reviewer 3



This submission describes a model for general and efficient style transfer. It is general in the sense that it does not need training for each style image that one would want to apply (unlike [26]), and it is efficient because it does not require solving an optimization problem as the initial work of Gatsby et al. [9]. The main idea is to train a deep decoder network, so that from any given layer in a VGG, one can generate the most likely image. This decoder is not trained as a classical autoencoder. Instead, given a fixed supervised VGG, an decoder counterpart of a VGG is trained so as to reconstruct the image. Given this trained decoder, the authors propose to transfer style by taking a given layer, make the covariance of the activations the identity and in a second step, make them follow the covariance of the style image at the same layer (called coloring). Given these new activations the desired style-transfered image is obtained by applying the corresponding decoder. Pros: - The proposed approach is very simple and intuitive. It does not require fancy losses, and uses simple baseline techniques to approximately match the distributions of activations. - This paper exhibits the very large expressive power of CNNs. Given a well trained decoder, simple linear operations in the feature space give meaningful outputs. - The proposed Coarse to Fine approach leads to very good visual performance. - The proposed algorithm is fairly efficient as for each new stye, it only requires computing several singular value decompositions (two for each layer, one for the image, one for the style). - The proposed model only has one parameter, namely the tradeoff between style and content. This is a great advantage to say [26]. - Overall the paper reads well and is easy to follow. Cons: - The description of the Decoder training could be made more clear by introducing the fixed encoder, let's say \phi. Then the training of the decoder could be written as the minimization w.r.t. a decoder \psi of the loss. That would avoid using strange notations including I_\input I_\output (which actually depends on the function that is optimized over...). - The formal description of the whitening and coloring could be made more practical by introducing the SVD of the activations (instead of using the eigenvalue decomposition of the covariance matrix). Questions: - To a large extent, this approach relies on the training of a good decoder, they quality of which strongly depends on the choice of data used for training. It would be interesting to see more results with worse decoders, trained on less data (or images with different statistics). - The model assumes that the "support" of the decoder (subset of the input space that was covered by the inputs) contains the activations of the style images. Provided that this is true and that the decoder is well regularized, the output should be correct. Would it be possible to assess to what degree does a dataset such as COCO cover well this space? Otherwise, the decoder might have ill-defined behaviors for parts of the input space that it has never seen during training. Overall, I think this submission in interesting and shows good results with a simple approach. I lean towards acceptance.

Reviewer 4



This paper proposes an alternative to the pixelwise content loss and Gram matrix-based style loss introduced by Gatys et al. After learning to invert a pre-trained classifier, style transfer is achieved by feeding both the content and style images through the classifier up to a certain depth, matching the content feature statistics to the style feature statistics using whitening and coloring transforms, and decoding the resulting features back to input space. The paper is well-written and the problem tackled is very relevant. The idea proposed is simple but appears to be very effective. It exhibits the very nice property that once the decoder is trained, style transfer is achieved without any expensive optimization loop. Have the authors tried the method using images of very different sizes than the ones used during training of the decoder? If so, did the reconstruction quality (and therefore the stylization quality) suffer? An issue I have with the evaluation (which is not unique to that paper) is that it relies on qualitative assessment, which is highly subjective. This is still an open issue for artistic style transfer in general, and because of the novelty of the proposed technique I am not weighting this issue too heavily in my judgement. Missing from the Related Work section is Ghiasi, Golnaz, et al. "Exploring the structure of a real-time, arbitrary neural artistic stylization network." To appear in BMVC2017. which extends work on conditional instance normalization by Dumoulin et al. and represents an approach concurrent to that of Huang et al. for extending style transfer to unseen styles.